# Chitosan-Based Nanoparticles of Targeted Drug Delivery System in Breast Cancer Treatment

**DOI:** 10.3390/polym13111717

**Published:** 2021-05-24

**Authors:** Yedi Herdiana, Nasrul Wathoni, Shaharum Shamsuddin, I Made Joni, Muchtaridi Muchtaridi

**Affiliations:** 1Department of Pharmaceutics and Pharmaceutical Technology, Faculty of Pharmacy, Universitas Padjadjaran, Sumedang 45363, Indonesia; y.herdiana@unpad.ac.id (Y.H.); nasrul@unpad.ac.id (N.W.); 2School of Health Sciences, Universiti Sains Malaysia, Kubang Kerian, Kelantan 16150, Malaysia; shaharum1@usm.my; 3Nanobiotech Research Initiative, Institute for Research in Molecular Medicine (INFORMM), USM, Penang 11800, Malaysia; 4USM-RIKEN Interdisciplinary Collaboration on Advanced Sciences (URICAS), USM, Penang 11800, Malaysia; 5Departement of Physics, Faculty of Mathematics and Natural Sciences, Universitas Padjadjaran, Jl. Raya Bandung Sumedang KM.21 Jatinangor, Sumedang 45363, Indonesia; imadejoni@phys.unpad.ac.id; 6Functional Nano Powder University Center of Excellence, Universitas Padjadjaran, Sumedang 45363, Indonesia; 7Department of Pharmaceutical Analysis and Medicinal Chemistry, Faculty of Pharmacy, Universitas Padjadjaran, Sumedang 45363, Indonesia

**Keywords:** breast cancer, chitosan-based nanoparticles, active targeting, EPR

## Abstract

Breast cancer remains one of the world’s most dangerous diseases because of the difficulty of finding cost-effective and specific targets for effective and efficient treatment methods. The biodegradability and biocompatibility properties of chitosan-based nanoparticles (ChNPs) have good prospects for targeted drug delivery systems. ChNPs can transfer various antitumor drugs to targeted sites via passive and active targeting pathways. The modification of ChNPs has attracted the researcher to the loading of drugs to targeted cancer cells. The objective of our review was to summarize and discuss the modification in ChNPs in delivering anticancer drugs against breast cancer cells from published papers recorded in Scopus, PubMed, and Google Scholar. In order to improve cellular uptake, drug accumulation, cytotoxicity, and selectivity, we examined different kinds of modification of ChNPs. Notably, these forms of ChNPs use the characteristics of the enhanced permeability and retention (EPR) effect as a proper parameter and different biological ligands, such as proteins, peptides, monoclonal antibodies, and small particles. In addition, as a targeted delivery system, ChNPs provided and significantly improved the delivery of drugs into specific breast cancer cells (MDA-MB-231, 4T1 cells, SK-BR-3, MCF-7, T47D). In conclusion, a promising technique is presented for increasing the efficacy, selectivity, and effectiveness of candidate drug carriers in the treatment of breast cancer.

## 1. Introduction

Breast cancer is the most common, chronic, and common causes of invasive and death in women worldwide [1,2,3], in both developing and advanced countries [4]. Breast cells become irregular and expand uncontrollably as breast cancer progresses, leading to a tumor’s development [5]. The lack of available treatment options for advanced cancer stages is the primary explanation for the highest cancer mortality rate [6]. Different therapeutic methods have been licensed for the treatment of this disease regardless of the stage of the disease, but the value of specific conventional techniques reduces due to the dynamic and heterogeneous nature of the cancer tissue [1].

Tumor eradication surgery followed by radiation and chemotherapy were conventional methods of treatment that often caused many harmful side effects on healthy cells [6,7,8]. All existing methods of BC treatment are not as effective as anticipated and are concerned with low drug solubility, low bioavailability, gastrointestinal permeability, low cellular uptake, emerging resistance, and adverse toxicity [2]. Several studies have been conducted to find a treatment for breast cancer, but there is no ultimate solution. There is a high recurrence rate for most treatments, so the search for effective anticancer agents continues [9].

Drug design and development uses various nanotechnologies to promote cancer treatments’ underlying shortcomings, known as cancer nanotechnology, for more therapeutically efficient and safer medicines [10]. In developing a new pharmaceutical carrier and delivery system, nanotechnology has garnered tremendous attention [11]. Nanocarriers will carry drug molecules to specific locations or targets and are expected to kill cancer cells without damaging normal cells [12,13]. There have been various nanocarrier innovations, such as polymer nanoparticles, liposomes, carbon nanotubes, dendrimers, and superparamagnetic nanoparticles [14].

During breast cancer therapy, many challenges are facing conventional delivery, such as:Physicochemical drug characteristics such as poor solubility, leading to poor pharmacokinetic characteristics.Insufficient specificity for breast cancer.Cellular-level and tumor microenvironment drug resistance.Difficulties of eliminating cancer stem cells.Inefficient drugs exposure to the metastatic locations [8,10,15].

These challenges can be overcome by:Technologies to strengthen the solubility and stability of anticancer drugs [16].Passive and active targeting is used by nano-based modalities to selectively target malignant tissues/cells [17].The distribution of various drugs helps decrease drug resistance.The nano-based vehicles have controlled release techniques for a medication.Drug efflux pathways can be blocked by nano-based vehicles.Usage of nanoformulation dependent on stimulus-responsive.Delivery of different medications and, thus, aims to minimize drug resistance [8,15].

Nanoparticles that use a natural polymer, chitosan, are efficient, cost-effective, and environmentally friendly nanoparticles among polymeric nanoparticles [18,19]. The cationic properties, electrostatic interactions, and biodegradability of chitosan nanoparticles (ChNPs) have attracted attention [20]. Biocompatibility is one of the most important criteria in biomaterials selection. ChNPs can be administered orally and intravenously into host cells [21]. ChNPs are used as a drug delivery system targeted at cancer therapy [6]. The search for target molecules that can regulate signal transduction has recently emerged as a globally popular biomedical research field [22].

Chitosan is efficient, cost-effective, and environmentally friendly sources of nanocarriers among the polymeric nanoparticles. Chitosan is a biological macromolecule with a wide range of bioremediation, anticancer properties, and carrier of drugs. The cationic properties, electrostatic characteristic, and biodegradability of ChNPs have drawn attention to the ease of oral and intravenous administration of carrier drugs into host cells. They consider the researchers’ overall positive views on the use of ChNPs through a targeted drug delivery system as a possible cancer therapy [6]. The search for targeted molecules that can impact signal transduction has evolved rapidly in biomedical science as a prominent field [22].

These studies indicate that nanoparticles based on chitosan could explore the combination of the EPR effect (passive target) and the target ligand (active target). Because of its additive or synergistic impact, the combination can achieve better therapeutic efficacy than single-modality therapy.

## 2. Nanotechnology in Drug Targeted Delivery System in Cancer Therapy

The use of nanotechnology shows rapid development in cancer therapy [23]. Nanotechnology can deliver drug molecules to target sites without damaging healthy cells [16,24,25]. Nanomedicine improves stability, solubility, drug half-life, the bioavailability of many chemotherapeutic drugs. Nanomedicine significantly reduced the peak drug concentration (Cmax), increased drug accumulation at target sites and the area under the curve (AUC) [10]. Typical small size and unique nanoparticle coating facilitate the delivery of hydrophobic anticancer drugs with decreased opsonization by the immune system to target locations in the body [2].

Through increased permeability and retention, NPs can boost drug accumulation in cancer cells (EPR). Ultimately, the combination of NPs–anticancer drugs can improve the therapy’s effectiveness by minimizing side effects by target-ligands interaction [8].

Various types of targeted DDS NPs use for BC investigations (Figure 1). NPs can be categorized into mesoporous silica, liposomal, polymer, metal, carbon, and protein-based NPs [8].

The success of NPs in the delivery of drugs varies for different properties, such as shape, scale, composition, and other structural aspects. NPs have a multi-component architecture based on diagnostic and/or therapeutic goals [2].

There are many strategies used by NPs to influence cancer cells:Protected drugs against hepatic inactivation, enzymatic degradation, and rapid clearance [26].It has enhanced cancer cell internalization by functionalizing with ligands over-expressed in tumor cells recognizable by receptors (targets) [2].It is influenced by cancer growth by controlling the tumor microenvironment (TME) [27]. Polymers can respond to and perceive exogenous stimulus (light, temperature, ultrasound, electrochemical triggers) or microenvironmental tumor (pH, enzyme activity, redox properties) to cause drug release to overcome these barriers [28].Escaped the multiple drug resistance (MDR) efflux transporters [2].Reduced the incidence and intensity of side effects [29].Carried contrast moieties contained allowing direct in vivo imaging (carrier visualization) [30].

Polymeric NPs (PNPs) have a larger deal with numerous properties for effective drug delivery and drug targeting. PNPs are easy to synthesize, inexpensive, biocompatible, biodegradable, non-toxic, non-immunogenic, and water-soluble [31]. PNPs can be either nanospheres or nanocapsules [32]. Natural polymer-based NPs are more beneficial in efficiency and targeted drug delivery than traditional drug delivery systems. The NPs were prepared using various natural polymers that have been shown to suppress the P-gp efflux pump [33].

PNPs can combine both therapy and imaging for controlled drug delivery, and provides drug protection and targeted drug delivery to improve the therapeutic index [23]. A schematic illustration of target-drug chitosan nanopolymer shown in Figure 2.

## 3. Biological Ligands for Nanoparticle Drug Delivery Systems

### 3.1. Passive Targeting

Nanoparticles (NP) can increase accumulation at the tumor cells through EPR effects [34]. In angiogenic tissues such as tumors, it is EPR that contributes to drug accumulation substantially. NPs that use passive targeting penetrate deeper into the vascular structures at the disease site, aided in part by sluggish lymphatic drainage. EPR is determined by the physicochemical properties of NPs, including size, shape (morphology), surface charge, and surface chemistry. These physicochemical properties can be easily modified by changing the component molecules or by the fabrication process [35].

Without a specific receptor target, nanocarriers can penetrate cancer cells through endocytosis and increase the number of drugs acting on the cells. Nanomedicine must have a diameter of d < 100 nm with a hydrophilic surface to avoid increasing drug targeting and improving drug circulation in the body. Their size has an influence on the amount and kinetics of nanomaterial accumulation at the tumour cells. The nanocarrier must be smaller than the cut-off proportion in the neovasculature, with the vehicle’s size acutely affecting extravasation to the tumor [35]. A passive mechanism is the accumulation of drugs at the tumour cells which involves a long circulation of drugs. Passive targeting is not a specific and competent drug delivery tactic [10].

### 3.2. Active Targeting

Many biologically significant ligands were identified and analyzed to facilitate the successful targeting of NPs. In addition, on the target cell membrane, these biological ligands attach to particular receptors. Ligand–receptor interactions will increase NP-containing drugs’ uptake and improve therapeutic efficacy [35]. Active targeting is used for tumor accuracy and delivery efficiency, requiring affinity-based identification, retention, and facilitated absorption of the target cells [15]. Pharmacologically active molecules (e.g., antibodies or monoclonal drugs) decorated with ligands on the nanocarrier will accumulate in the target cells [10].

For this purpose, various types of ligands have been used, including monoclonal antibodies, aptamers, and proteins, polysaccharides, nucleic acids, and small molecules [36]. In two general ways, NPs with this ligand function. Chemically conjugated or mechanically adsorbed to NP after the forming of NP, or they may bind to NP components before formation, such as polymers [35]. A variety of specific molecular interactions, such as receptor–ligand-based interactions, charge-based interactions, and facilitated motive-based interactions with substrate molecules, are the basis of chemical affinity for active targeting. Various biomolecules can attach ligands, including antibodies, proteins, nucleic acids, peptides, carbohydrates, and small organic molecules, as shown in Figure 3. Surface molecules expressed in diseased cells, proteins, sugars, or lipids contained in organs, molecules present in the microenvironment of the diseased cell (or secreted by tumor cells), and also the physicochemical environment surrounding them, may be the target substrate [15].

These characteristics should have an ideal active target moiety:The number of target moieties is higher in tumor cells than in normal tissues [37].It should also be measured at locations that nanocarriers can easily reach, such as superficial receptors rather than intracellular targets.In order to allow competent targeting, the concentration should be high enough.It must ideally associate its levels with malignant activities, whether drug resistance or active targets, to target these threatening tumors on a priority basis.Targeting can simplify procedures that facilitate the delivery of drugs [10].

## 4. Chitosan-Based Nanoparticles Preparation and Modification

Chitosan is a deacetylation process derived from chitin and is composed of d-glucosamine and N-acetyl-d-glucosamine units linked to β-(1→4) [38,39,40]. Chitosan is one of the possible drug carrier nanoparticles (NPs) because of its biodegradability and biocompatibility and displayed non- or minimal toxicity [41]. The cationic nature, which increased adhesion through electrostatic interaction to the negatively charged mucosal surface, is one of the most significant chitosan characteristics, resulting in improved drug internalization into targeted cells [38,42,43,44]. A significant barrier to its implementation is only soluble in an acidic medium. The extensive groups of aminos and hydroxyl serve as the target groups for chemical changes to improve solubility [38]. Multiple self-assembled copolymers and CS-based block hydrogels have been built. The biocompatibility, low immunogenicity and biodegradability of CS are good. Under the in vivo enzyme, CS will break down into water and carbon dioxide and become an endogenous species, ensuring no harmful effects from the products of degradation. Due to its increased solubility at slightly acidic pHs, such as those present in the microenvironment of the tumor, CS is generally used to develop pH-sensitive DDS [45].

### 4.1. Preparing Method Chitosan-Based Nanoparticles

For chitosan nanoparticle preparation, various methods have been described. A few of them, as shown in Figure 4, are as follows. The nanoparticles can be classified into three groups according to the difference in the preparation method.

#### 4.1.1. Self-Assembled

These nanoparticles are made from amphiphilic chitosan derivatives. In the water process, hydrophobic chains spontaneously create a reservoir for dissolved and slightly dissolved drugs. The hydrophilic chain functions around the core like a shell, which is exposed to the water phase. One of the most common derivatives used for the production of self-assembled nanoparticles is hydrophobically modified glycol chitosan [38].

#### 4.1.2. Ionic Cross-Linking

There are two types of ionic cross-linking: first, the manufacturing processes are simple does not use organic solvents or high temperatures, and no chemical interactions involved. These benefits make this procedure effective for thermolabile drug. Ionic cross-linking is carried out because of the interaction between positive charge of chitosan and negatively charged macromolecules or anionic cross-linking substances. A chitosan acid solution was prepared, together with stirring and sonication, and an ionic cross-linker was applied dropwise. Second, through chemical interactions between the cross-linking substance, the primary amine group, chitosan micro-/nanoparticle is formed. Glutaraldehyde, formaldehyde, vanillin, and genipin are common cross-linkers [46].

#### 4.1.3. Polyelectrolyte Complexes

A polyelectrolyte complex (PEC) is formed from a solution carrying two polyelectrolytes. The PEC formation is mainly due to the intense coulomb interaction among polyelectrolytes charged opposite to each other. The formation of the complex results in at least a partial neutralization of the polymer charge. Generally, the complex obtained will settle or escape from the solution to produce a rich and complex liquid process (coacervate). An important driving force for the formation of PEC is the increase in entropy caused by the release of these low molecular weight counter ions to the medium. While the formation of PEC is responsible for electrostatic interactions between the complementary ionic groups of polyelectrolytes, hydrogen bonding, and hydrophobic interactions also contribute to complexity. A mixture of structures, such as irregular scrambled eggs and highly organized ladder-like organization, can be seen as a chain arrangement in PEC. The actual structure of the hydrophobic and hydrophilic regions therefore makes PEC a particular class of cross-linked hydrogels that are physically susceptible to pH and other environmental factors, such as temperature and ionic strength [47].

### 4.2. Chemical Modifications or Incorporation

Chemical modification or integration with functional materials must produce a better and stable release in vivo [38].

#### 4.2.1. Chitosan Derivatives Nanoparticles

Chitosan can be modified chemically because it has a hydroxyl, acetamide, and amine functional group. The chemical modification will not change the basic framework of chitosan and maintain the original physicochemical and biochemical properties while introducing new or enhanced properties [48]. Trimethylated chitosan has high water solubility. These derivatives enhance the complexing between chitosan and nucleic acids [49]. Chitosan glycol thiolate produces a stable complex through electrostatic interactions and disulfide cross-linking [50]. Galactosylated trimethyl chitosan-cysteine/siRNA complexes affect the affinity of siRNA binding on antitumor efficacy [46]. There is an opposite effect of this binding. The strong binding force forms the serum complex’s stability but the inefficient release of siRNA within the cell [51]. The authors propose that the carrier and siRNA’s binding strength should be adjusted to increase efficiency and antitumor activity [38].

#### 4.2.2. Conjugates/Complexes of Chitosan-Polymer Nanoparticles

C3-OH, C6-OH, C2-NH2, and acetyl amine and glycosides are functional groups on the chitosan molecule. Acetyl-amino bonds, which are not easily broken, are just as stable as glycosidic bonds. C6-OH with a small steric hindrance is a primary hydroxyl group, and C3-OH with a large steric hindrance is a secondary hydroxyl group [47]. In the chitosan molecule, the groups of C6-OH and C2-NH2 can be utilized to add other groups with different types of molecular designs. Chitosan chemical modification can enhance its physical and chemical properties and expand the application and associated fields of study [52]. The functional groups of chitosan chemical modification are shown in Table 1.

## 5. Chitosan-Based Systems as Drug Carriers in Targetted Drug Delivery Systems in BC Treatment

The four major benefits of such approaches to nanotechnology are helping solubilize hydrophobic drugs, prolonged drug in system circulation, it can passively target tumour cells through the EPR effect, allow stimulus-responsive materials to ensure provide drug at the target cells, and functionalized targeting ligands to achieve specific tumor cell binding and uptake, thus minimizing off-target side effects [45].

Breast cancer and other proliferating or developing cells have specific characteristics:The glucose uptake rate increases dramatically, and lactate is generated, even in the presence of oxygen and fully functioning mitochondria [53].pH gradients between the normal tissue (blood physiological pH 7.4) and the tumor site (tumor extracellular pH 6.5) [28,45,54].Hyperpyrexia (temperatures raised to 40–42 °C due to the increasing rate of glycolysis and rapid cell proliferation) is also characterized by the tumor microenvironment [45].The concentration of intracellular glutathione (GSH), a tripeptide responsible for reducing disulfide bonds, is about 2–3 orders higher than the extracellular GSHH order [55].Positive charges on the surface of nanoparticles are also necessary for transfection into the cells because of the negative charge on membranous cells [56].The concept of developing targeted anticancer therapeutics has led to the development of breast cancer receptor type 2 (HER-2), estrogen receptor (ER), up-regulated with human epidermal growth factor receptor [57].In breast cancer cells, the folate receptors (FRs) are over-expressed [58].

The encapsulation technology in nanopolymers is safe, less toxic, and effective alternative therapy. Using encapsulation or direct attachment, anticancer drugs can be loaded via ChNPs [6]. This method is influenced by chitosan’s structural characteristics, such as molecular weight (Mw), degree of deacetylation (DD), and substitution place. These physicochemical properties have a major impact on the antitumor effectiveness of chitosan-based nanoparticles. These variables are not different but have a cumulative effect on the nanoparticles. The priority issue for rationally engineered nanoparticles is chitosan’s acceptable choice, as shown in Figure 5 [38].

Cellular uptake and zeta potential were decreased with the decrease of Mw and DD [38]. With the rising amount of chitosan, the surface charge increased [59,60,61]. Chitosan’s beneficial properties are LMW chitosan, a high DA, a diameter of 100 nm and a high zeta potential [62]. High DD generates high surface charge density nanoparticles, resulting in improved cell absorption and antitumor efficacy [63]. In general, the binding of one ligand molecule makes it easier to bind the resulting molecules by cooperative effects, jointly increasing the binding efficiency and subsequent behavior [15].

Nanocarriers can be used for both passive and active targeting methods. Increased permeability and retention effect (EPR), nanoparticle size, and unique properties enhance cytotoxicity. ChNPs with anticancer drugs penetrate tumor cells through the gaps in the effect of EPR on tumor angiogenic blood vessels (600–800 nm) [64]. ChNPs can be used in EPR and active targeting [6]. Table 2 shows that the effect of EPR/passive targeting seen from the size, zeta potential, and shape of the nanoparticles will impact cellular uptake. Most manufacturing methods produce NPs below 200 nm. NPs are avoided by renal filtration through the glomerular capillary walls (<10 nm) and are not reabsorbed. Nanocarriers (50–200 nm) cannot exit the continuous blood capillaries in their intact form, thereby reducing conjugate excretion [65].

Through the mechanisms of phagocytosis, micropinocytosis, endocytosis, direct diffusion, or adhesive interactions, nanoparticles can enter cells. The size and shape affect the uptake of nanoparticles in cells. How cells internalize nanoparticles is determined by their physical and chemical characteristics, such as size, shape, surface charge, and composition. Any size greater than 200 nm, facilitated by adding a ligand such as folic acid or epithelial protein, will increase endocytic macrophages. It is known that size is an important factor. In previous studies, smaller NPs were more likely to be imported into cells via endocytosis or diffusion, and larger NPs were more likely to be imported into cells via phagocytosis [77]. Free energy that results from ligand–receptor interaction and receptor diffusion kinetics onto the wrapping sites on the cellular membrane are two factors that prescribe how fast and how many nanoparticles enter the cellular compartment via wrapping [78].

In Table 2, CNP and its derivatives show a positive zeta potential. The potential of the zeta plays a significant role. Positively charged NPs prefer to be internalized rapidly and intensely by negatively charged cancer cells because of electrostatic interactions [34]. The presence of other groups of drug compounds can change the charge, as shown in Table 2. NPs with positive charges, however, can interact with components of blood circulation that are negatively charged. NPs are rapidly captured and subsequently removed by the liver through the reticuloendothelial system (RES), resulting in loss of pharmacological activity and decreased accumulation of cancer cells. Li et al. using chitosan-lipoic acid NPs grafted with histidine to form HCSL-NPs. HCSL-NPs will be negatively charged in the blood circulation and become positively charged on TME. PH gradient between normal tissues (physiological blood pH 7.4) at the tumor site (Tumor extracellular pH 6.5). By breaking the amide link or protonation of amino groups and imidazole groups, the phenomenon of charge conversion can be accomplished [34].

In Table 2, chitosan and its derivatives can be used as carriers for breast cancer therapy drugs. The challenges of NPs in breast cancer therapy are the time of drug circulation, increased cell uptake, increased cytotoxicity, selectivity to normal cells, and decreased tumor size in vivo. The correlation of this challenge is that the increase in cell uptake will increase the drug concentration in the cell. Drug concentration in cancer cells will increase cytotoxicity. Increased cytotoxicity is a measure of the effectiveness and efficiency of NPs. All that can be done by optimizing passive targeting and active targeting.

NP chitosan can bind CPP. CPP is defined at physiological pH as a short peptide, water-soluble and partially hydrophobic, and/or polybasic (at most 30–35 amino acid residues) with a positive charge. The main feature of CPPs is that, without chiral receptors and without causing major membrane damage, they can penetrate cell membranes at low micromolar concentrations in vivo and in vitro. CPP leads to increased cellular uptake [72]. An increase will follow the increase in cellular absorption in cytotoxicity.

Due to its hydroxyl, acetamide, and amine functional groups, chitosan is ideal for chemical changes, providing a type of derivative that has more benefits than free chitosan. Modification of CS NP will be able to avoid macrophages in the reticuloendothelial system (RES). GCS is a chitosan derivative with an ethylene glycol group, replacing specific polysaccharide repeating units with a hydroxyl group for better hydrophilicity [65]. Reduced clearance (escaped phagocytosis of RES cells) will increase drug circulation time, tumor aggregation, and toxicity [79,80].

External and internal stimulus are used to release cancer drugs from the chitosan polymer. This environmental change causes stimulation to release drugs against cancer cells [6]. Cs can be incorporated with compounds, which can be activated by external stimulus.

ChNP can induce cell apoptosis by interfering with cell metabolism and tumor-induced cell growth. Previous research on ChNPs found that low molecular weight chitosan and chitooligosaccharide can induce tumor cells. Adsorption of blood plasma and other tissues will speed up the identification of the reticuloendothelial because it is considered foreign to human antibodies. Macrophages will ingest ChNP and excrete it through the body’s circulation. The surface charge of ChNP adsorbs plasma proteins, which allows macrophages to identify them quickly. This process helps the rapid penetration or entry of the drug tumor tissue. ChNP is readily absorbed with high surface potential, high polarity, and amphipathic or hydrophilic properties and remains in the tissue longer [6]. Cationic polymers provide siRNA protection and enhance endosomal siRNA release and cytosol delivery [45].

Although nanoparticles have been used to fight cancer, the main problem is the large number of nanoparticles that accumulate after treatment, so it is necessary to look for nanoparticles with a minimal dose. Therefore, ChNPs have been introduced, which play a promising role as nanocarriers for chemotherapy drug delivery, as shown in Figure 6. Anticancer drugs are blended with ChNPs with other hydrophobic surface coating agents to make the nanocarrier sustainable until it hits the tumour cells [6].

It is well established that the tumor’s microenvironment is slightly acidic (pH 6.0–7.0) and that the endosome/lysosome is seriously acidic (pH 4.5–5.5). Under normal physiological conditions, the negative-to-positive charge reversal behavior triggered by pH can ensure the stability of ChNPs and promote their cellular uptake and endolysosomal escape. The pH-sensitive surface charge reversal behavior of the cationic polymer and anionic polymer nanoparticles can prolong blood circulation time, reduce side effects, and improve cell uptake [81].

Photoacoustic contrast agent for cancer cell imaging through the use of GC. GC is a derivative of chitosan with ethylene glycol moieties, substituting hydroxyl groups for improved hydrophilicity for some polysaccharide repeating units. GC can serve as a multipurpose imaging and therapeutic agent because amine groups in the repeating unit provide chemical modification sites [65]. The preparation and in vivo process for the simulated photodynamic and photothermal therapy of a multifunctional nanosystem against breast cancer is described [38].

The number of cell lines widely used for breast cancer studies is extremely small, with cell lines such as MCF7, T47D, and MDAMB231 accounting for more than two-thirds of cell lines used in the associated studies [82]. MCF-7 cells represent a very important candidate as they are used ubiquitously in research for estrogen receptor (ER)-positive breast cancer cell experiments [83]. MDA-MB-231 is a highly aggressive, invasive, and poorly differentiated triple-negative breast cancer (TNBC) cell line as it lacks oestrogen receptor (ER) and progesterone receptor (PR) expression, as well as HER2 (human epidermal growth factor receptor 2) amplification [84]. 4T1 cells share substantial molecular features with human TNBC. As 4T1 is a common model for metastatic tumors, our data supports the rational design of mode-of-action studies for pre-clinical evaluation of targeted immunotherapies [85]. SK-BR-3 is a human breast cancer cell line that overexpresses the Her2 (Neu/ErbB-2) gene product [86].

ChNPs can work by delivering anticancer drug to all widely use cancer cells (Table 2) that represent all type breast cancer ER (estrogen receptor), PR (progesterone receptor), Her2 (human epidermal receptor 2), and triple negative breast cancer.

## 6. Stimuli-Responsive Materials Based on Chitosan-Based Nanoparticles

While targeted drug delivery systems with a controlled release profile have recently developed, certain restrictions such as premature and low release of chemical agents into the target cell of cancer can limit the in vivo impacts of such modalities [87,88]. Several advanced smart drug delivery systems have been developed to provide a stimulus-triggered on/off drug release in response to various extra/intra stimulus in an on-demand manner [88,89]. The presence of functional groups in the polymeric matrix of chitosan is responsible for this ability, which makes it an exciting candidate for smart nanocarriers. In order to gain both passive and activated targeting mechanisms and simulcast endogenous or exogenous stimuli, the CS-based targeting NSs can be configured. CS-based targeted nanocarrier with the capability of controlled-release of cargo drug molecules in response to specific stimulus, pH [90,91,92,93], redox gradients [94,95], ultrasound [96], light, temperature [97], magnetic [98], and dual stimuli [99].

## 7. Perspective

ChNPs technology represents different modifications in drug development to increase the accumulation of drugs in cancer cells [15,63,100,101,102], cellular uptake [103,104], cytotoxicity [105,106,107,108], and selectivity to normal cells [87,109,110] in vitro and in vivo cells. By considering the EPR effects and the active targeting portion, chitosan-based nanoparticles will attain all these objectives. ChNPs can work by delivering anticancer drug to all widely use cancer cells that repre-sent all type breast cancer ER (estrogen receptor), PR (progesterone receptor), and Her2 (human epidermal receptor 2) and triple negative breast cancer. Our objective research showed that the chitosan-based nanoparticle technology synergizes the overall impact of EPR effects and the active targeting component in deliver anticancer drug againt breast cancers cells. Firstly, decreasing the size of NPs enhances the solubility and stability of drugs [109,111,112,113,114]. and the surface charge of NPs, improves the protection of drugs in blood circulation [115,116,117], and increases the uptake of drugs in cancer cells [104,115,118]. Secondly, the incorporation of the biological ligand can vectorize NPs to specific cancer cells. It will increase drug accumulation in cancer cells, cellular uptake, and cytotoxicity. On the other hand, the use of chitosan as a natural polymer inhibits efflux P-gP pumps [33,119,120]. The modification abundant functional group of chitosan will give a wide range of modifications [46,52,121]. This modification chitosan will maximaze other cancer cells’ spesial characteristics such as pH gradient [122,123,124,125], temperature gradient [126], and redox [127,128,129]. Chitosan will use exogenous stimuli at spesial site to monitor release [127,128,129] and carry a contrast photoacoustic agent to diagnose cancer [127,128,129]. Therefore, chitosan-based nanoparticle formulations represent one option to resolve these limitations: poor water solubility profile and its low selectivity of the anticancer drug and delivery to the target cells.

## 8. Conclusions

Therapy using ChNPs can work by delivering anticancer drugs to all widely used cancer cells, increasing cytotoxicity and improving drug accumulation, selectivity, and efficacy. It is possible to integrate various hydrophilic or hydrophobic chemotherapy drugs, nucleic acids, and photosensitizers into chitosan-based nanoparticles and deliver them through EPR effects and ligand–receptor interactions to specific tumor cells. Furthermore, chitosan’s cationic structure enables the effective release into the cytoplasm of nanoparticle endosomes. Based on cancer cells’ characteristics, such as pH gradient, temperature gradient, load, and cancer cell metabolism, chitosan can be used. ChNPS can overcome restrictions such as premature and low release of chemical agents into the target cell of cancer can limit the in vivo impacts of such modalities, by providing a stimulus-triggered on/off drug release in response to various extra/intra stimulus in an on-demand manner (stimulus pH, redox gradients, ultrasound, light, temperature, magnetic, and dual stimuli). ChNP offers many opportunities for breast cancer therapy.

## Figures and Tables

**Figure 1 polymers-13-01717-f001:**
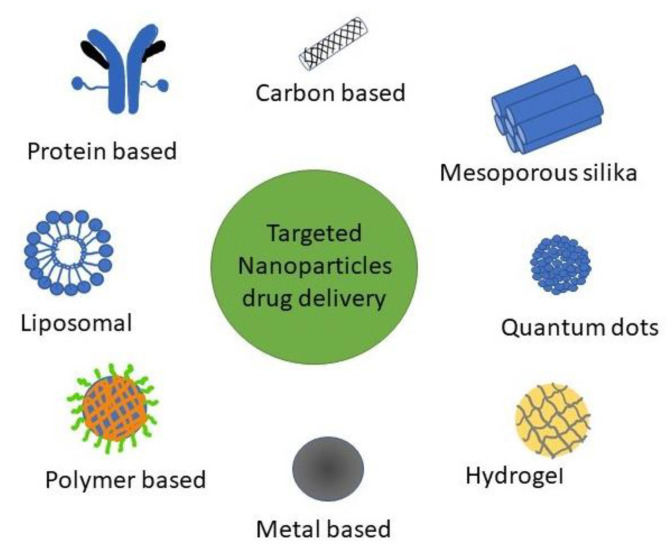
Types of NPs for targeted DDS used in BC research.

**Figure 2 polymers-13-01717-f002:**
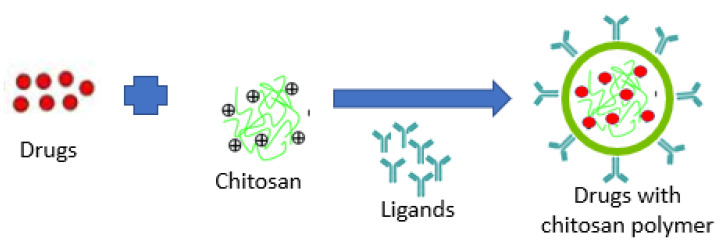
Schematic illustration of target-drug chitosan nanopolymer.

**Figure 3 polymers-13-01717-f003:**
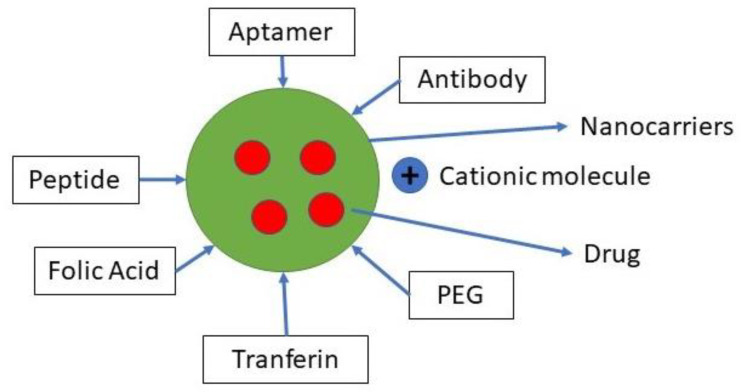
Ligan for active targeting.

**Figure 4 polymers-13-01717-f004:**
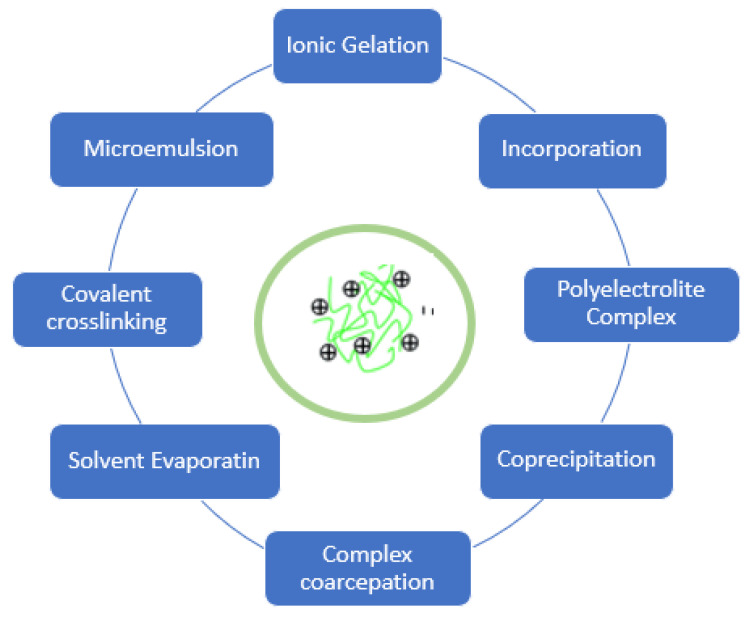
Preparing method of chitosan nanoparticles.

**Figure 5 polymers-13-01717-f005:**
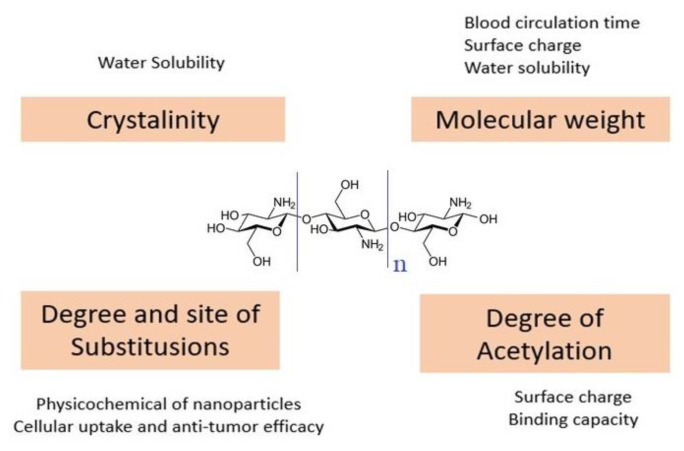
Chitosan structural features and their impacts on chitosan-based nanoparticles.

**Figure 6 polymers-13-01717-f006:**
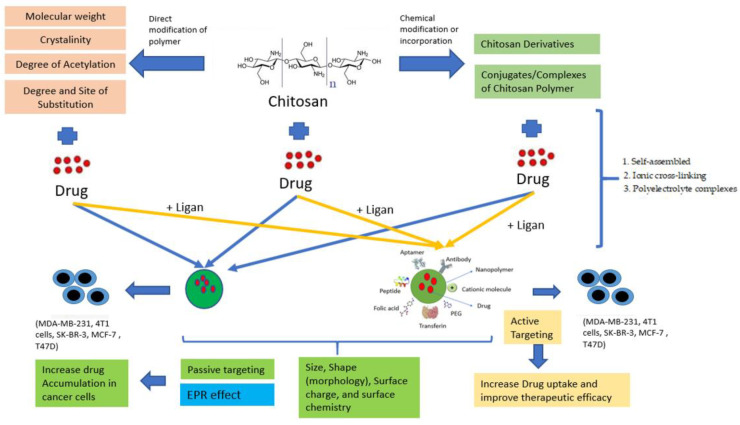
Passive and Active Targeting in ChNPs.

**Table 1 polymers-13-01717-t001:** Functional groups of chitosan chemical modification.

No	Functional Groups	Chemical Reaction
1	C2–NH2	Metal coordination Chemical coupling Chemical cross-link Graft copolymer Alkylation Acylation Shift Base
2	C3–OH	Sulfonation Alkanoylation
3	C6–OH	Metal coordination Chemical coupling Chemical cross-link Graft copolymer Esterification Carboxymethylation Alkylation
4	Acetyl amino	
5	Glycosidic bond	Glycosidic bond cleavage degradation

**Table 2 polymers-13-01717-t002:** Studies on Chitosan-based Nanoparticles of Targeted Drug Delivery Systems.

Chitosan Composition	Agents of Drugs	Preparing Method	EPR Effect	Activ Targeting Receptor	BC Cell Line	Effect	Reference
Carboxymethyl dextran (CMD) Chitosan nanoparticles (ChNPs)	Co-delivery of IL17RB siRNA and DOX	The Ionic gelation Method	d = 114 nm size, PDI = 0.3 and zeta potensial = 10.1 mV		MDA-MB361 cells	A significant silencing of NF-kB and Bcl-2 relative gene expression, apoptosis induction and migration inhibition	[66]
Glycol-chitosan-coated gold nanoparticles(GC-AuNPs)	AuNPs	Ethylene glycol moieties, substituting chitosan’s hydroxyl groups.	d = 94.4646.45 nm. zeta potensial 37.44.4 mV		MDA-MB-231	Enhanced cellular uptake, tumor accumulation, improved tumor-targeting of GC-AuNPs	[67]
Histidine-grafted chitosan-lipoic acid NPs (HCSL-NPs) were	doxorubicin (DOX)	Congujation	D = 106.0 nm, Z = −25.0. PDI = 0.129		4T1 cells	Enhanced internalization at extracellular pH, rapid release of intracellular drugs, and improved in vitro cytotoxicity against 4T1 cells were shown.	[34]
D-alpha-tocopherol polyethylene glycol 1000 succinate conjugated chitosan (TPGS-g-chitosan)	Docetaxel, Trastuzumab	Combined modified solvent evaporation technique with ionic cross-linking	186 nm 1.41 ± 0.20 mV	HER-2 receptor targetted	SK-BR-3	Cellular uptake and cytotoxicity have been enhanced. Increase in AUC and prolonged circulation of 1.4793 and 0.2847 μg/mL and greater safety than Docel TM.	[57]
Multifunctional hyaluronic acid/hydroxyethyl chitosan nanocomplexes	doxorubicin and 5-aminolevulinic acid.	Self-assembly method.	140 nm −24.6 mV, near-spherical shaped	The anti-HER2 antibody targeting moiety	MCF-7	Enhanced the cellular uptake, displayed pH-responsive surface charge reversal, and drug release.	[68]
Aldehyde hyaluronic acid (AHA) and hydroxyethyl chitosan (HECS)	Doxorubicin (DOX) and cisplatin	Conjugation and Self-assembly.	∼160 nm. −28 mV near- spherical morphology	HER2 receptor	MCF-7	The cellular uptake of the nanoplatforms was significantly improved by HER2 receptor-mediated active targeting. Improved stability.	[69]
Encapsulated O-succinyl chitosan graft Pluronic^®^ F127 (OCP) copolymer nanoparticles	Doxorubicin (DOX) with an anti-HER2 monoclonal antibody	Conjugated And Ionic cross-linking agents.	d = 34.92–48.79 nm	HER2 receptor	MCF-7	At pH 5.0.0, the drug was quickly and fully released from the nanoparticles. It improved cytotoxicity and selectivity. High efficiency of encapsulation.	[70]
Chitosan and pectin	Ribosome-inactivating protein (RIP)	polyelectrolytes complex and conjugation process with antiEpCAM antibody	376.8 nm + 36.05 mV with index polydispersity of 0.401	epithelial cell adhesion target	T47D and Vero cell lines	Increased the cytotoxicity of RIP. Low selectivity.	[71]
Low viscosity Chitosan and alginate	Ribosome-inactivating protein (RIP) from M. Jalapa L. leaves (RIP)	conjugated with anti-EpCAM antibody	D = 130.7 nm, +26.33 mV polydispersity index of 0.380	epithelial cell adhesion target	T47D Breast Cancer Cell Line	Enhance cytotoxicities, less selectivity.	[72]
The copolymer of chitosan and polyethylene glycol (PEG)	Superparamagnetic iron oxide nanoparticle (SPION) and fluorescent dye	A derivative of chitosan and Chitosan and polyethylene glycol (PEG) copolymer-coated conjugated SPIONs were labeled for optical detection and conjugated with a monoclonal anti-neu-receptor antibody (NP-neu).	Z = 44 and small PDI values	a monoclonal antibody at neu receptor (NP-neu).	mouse mammary carcinoma (MMC) cells	In MR images of primary breast tumors, significant contrast enhancement was provided, and high uptake was shown.	[73]
Folic acid-gallic acid-N, N, N-trimethyl chitosan (FA-GA-TMC)	cubic-like selenium nanoparticles (SeNPs)	Conjugation and the self-assembly method.	D = 300 nm, 30.1 mV.	Folic acid receptor	Cancer cells and normal cells (WI-38)	Improved anticancer efficacy and cellular uptake against breast cancer cells while demonstrating good selectivity.	[56]
Poly(N-vinylcaprolactam) (PNVCL)-chitosan (CS) nanoparticles (NPs).	doxorubicin (DOX) and cell-penetrating peptide	Conjugation and self-assembly	D = 120 ± 15 nm (n = 8). Z = −12.5 ± 2.5 mV. low polydispersity.	Cell-penetrating peptide (CPP)	MCF-7 TNB breast cancer cell line	Improved cytotoxicity, showing a selective reduction in tumor volume and prolongation of life span.	[45]
PEGylated chitosan and poly-L-arginine.	Superparamagnetic iron oxide nanoparticles (SPION) and siRNA	Conjugation and self-assembly	HD (nm) 213 ± 36 0.43 ± 0.05 Z = 30.7 ± 1.4	Cell-penetrating peptide (CPP)	MDA-MB-231 triple-negative breast cancer cells.	show a high uptake, the efficacy of the siRNA retention and protection, downregulation of GFP expression.	[74]
Oxidized hyaluronic acid-decorated dihydroxy phenyl/hydrazide bifunctionalized hydroxyethyl chitosan (DHHC)	gold nanorod (GNR) and Doxorubicin (DOX)	Congujation, chitosan derivates.	D = 94.0 nm, Z = +25.3 mV	Hyaluronic acid	MCF-7 cells	Enhanced cellular uptake and enhance cytotoxicity	[75]
Chitosan and alginate nanocapsules	Iron-saturated bovine lactoferrin (Fe-bLf)	Polyelectrolyte	Spherical size and d = 322 ± 27.2, z = −1.29 mV, PDI = 0.084	Low-density lipoprotein receptor and transferrin receptor	MDA-MB-231	Improved antitumor activity in breast cancer by internalizing and regulating micro-RNA expression via the low-density lipoprotein receptor and transferrin receptor.	[76]

## Data Availability

Not applicable.

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
