# Peer review of "Chitosan-Based Nanoparticles of Targeted Drug Delivery System in Breast Cancer Treatment"

_polymers, 2021, doi:10.3390/polym13111717_

Round 1
Reviewer 1 Report
The paper entitled "Chitosan-based Nanoparticles of Targeted Drug Delivery System in Breast Cancer Treatment" by Yedi Herdiana and co. is a literature review which discuss the modification in ChNPs for drug delivery for cancer treatment.
The subject is interesting, but the paper needs some improvements:
1) In abstract, the authors sustain that "The objective of our review was to summarize and discuss the modification in ChNPs from published papers recorded in Scopus, PubMed, and Google Scholar." Please insert more perecisely, what is the particuar scope of this paper, among many other published in this subject.
2) paragraph "2. Nanotechnology in drug targetted delivery system in cancer therapy" , page 3 row 114-117 : "NPs can be categorized into silica NPs that are liposomal, polymer, metal, carbon, protein-based, and mesoporous (8). "
Please give examples from literature for all these silica NPs and discuss the strategics of NPs to influence cancer cells in all cases.
3) Paragraph "3.1. Passive targeting" page 4, please give more examples of changing physicochemical proprieties from literature to sustain the text: "These physicochemical properties can be easily modified by changng the component molecules or by the fabrication process". What are the most cmmom proprieties which are changed and what implications have ?
3) Please introduce a paragraph with "Stimuli-responsive materials based on chitosan NPs" because nowdays this is a very important subject and those materials are the future for cancer therapy.
Reviewer 2 Report
This is such a good review, the authors collected the most interesting articles to write this paper. However, it lacks some descriptives and should revise some sections more specifically to focus on breast cancer. I advice to revise this paper based on below comments:
The authors first need to explain how this review paper brings new information to the readers because there are plenty of Review papers already published Elsevier, Except Breast cancer application.
For Ex.
1) Advances in chitosan-based nanoparticles for oncotherapy. https://doi.org/10.1016/j.carbpol.2019.115004
2) Tumor targeting strategies for chitosan-based nanoparticles.https://doi.org/10.1016/j.colsurfb.2016.09.020
3) Chitosan-based nanoparticles for tumor-targeted drug delivery. https://doi.org/10.1016/j.ijbiomac.2014.10.052
4) Chitosan-based nanoparticles as drug delivery systems: a review on two decades of research. https://doi.org/10.1080/1061186X.2018.1512112
5) Chitosan-based nanoparticles: An overview of biomedical applications and its preparation. https://doi.org/10.1016/j.jddst.2018.10.022
29 ChNps use the characteristics: Change to ChNPs. Keep consistent wording.
29 of the EPR: Explain EPR
32 of drugs into specific cancer cells. : Give specific details? What type of cancer cells?
The abstract is written in a general manner, not specific to breast cancer (BC) treatment. Focus on why ChNPs are unavoidable material in BC, what is the impact, Mechanism/mode of action and recent advancements in BC and how this review specifically collects/focuses on ChNPs in breast cancer?
Include the reference published by Vahideh Alinejad et al. Co-delivery of IL17RB siRNA and doxorubicin by chitosan-based nanoparticles for enhanced anticancer efficacy in breast cancer cells. https://doi.org/10.1016/j.biopha.2016.06.037
Figure 1. Did you draw all the images by yourself or got them from previous papers? If so need copyright permission.
Provide the possible Mechanism of action of ChNPs in Breast cancer treatment as one Figure (Schematic Illustration). Also, include Signaling pathways and molecular/receptor interaction of ChNPs in different Breast cancer cells as one Figure. Also, provide the changes of structural and functional groups in Chitosan on chemical modification and how modifying chemical structure improves the anti-cancer activity/ which functional group of chitosan regulates anti-cancer activity as one Figure. All the above Figures will definitely give significant value to this review and attract more readers.
Perspectives
415 Our objective research showed...anticancer drug and deliver to the target cells.: Again not at all related to Breast cancer as shown in the Title.
Table 2. Studies on Chitosan-based Nanoparticles of Targeted Drug Delivery Systems.: Describe the available actual mechanism of the ChNPs on the cancer cells listed in Table 2.
Table 2 Legend Activ : Active
Conclusion: Not at all related to or focused on breast cancer and ChNPs. This section was also written in a general way. Describing very basic statements regarding the use of ChNPs in cancer treatment, which was already published in several review papers.
By reading the title, this review seems to describe more specifically the recent advancement of Breast cancer and ChNPs treatment. But the perspectives and conclusion did not disclose the above concept (simply describing some general statement). Revise.
Round 2
Reviewer 1 Report
The authors made the changes and the paper can be accepted in present form
Author Response
The authors made the changes and the paper can be accepted in present form
Thank for reviewing this manuscript, Thank you for reviewing this manuscript, thank you for the input which will make the manuscript more high quality.

Reviewer 2 Report
The revised version is much improved, however not convincing and satisfactory.
Most of the images were included without proper permission. Do the necessary amendments to avoid any scientific conflict.
1.3 out of 5 publications mentioned are included in the bibliography, I use them as a reference source. The difference with the mentioned publications is the focus on breast cancer. The publication describes cancer cells in general.
New comment: However, in both cases, the anticancer activity of chitosan was described and the mechanism of chitosan is same for all cancer cells? Or any unique pathways involved in breast cancer for Chitosan?. If the authors want to claim specifically breast cancer, then explain the unique features and mechanism of Chitosan against breast cancer not general anti-cancer mechanism.
Figure 1. Did you draw all the images by yourself or got them from previous papers? If so need copyright permission.
Done, I draw all image by myself.
New comment: This is wrong. Please respond correctly. The all the images were not drawn by yourself. Some of them were taken from the Internet. For example.
The image Quantum dot was copied from QNA Technology website. https://qnatechnology.com/technology/what-are-quantum-dots/?lang=en
The image mesoporous silica was copied from Figure 1 of the published book by Adebola Iyabode Akinjokun et al. DOI: 10.5772/63463
Figure 3. The Aptamer image was copied from Figure 1 of published work by Olga Wolter et al. DOI: https://doi.org/10.1523/JNEUROSCI.1969-16.2017
Transferin image was copied from online https://pdb101.rcsb.org/motm/35. It seems all the images folic acid, peptide, antibody, and PEG were taken from external sources. Please acknowledge the correspondence and give copyright permission to reuse these images, otherwise after publish the authors should face the consequences.
Figure 3 is very poor resolution and not clear.
Figure 5 and 6. The image chitosan was copied from External website (https://www.indiamart.com/proddetail/chitosan-low-density-11290818962.html)
The image MDA-MB-231 of Fig 6 was also copied from an external website.
- Provide the possible Mechanism of action of ChNPs in Breast cancer treatment as one Figure (Schematic Illustration). Also, include Signaling pathways and molecular/receptor interaction of ChNPs in different Breast cancer cells as one Figure. Also, provide the changes of structural and functional groups in Chitosan on chemical modification and how modifying chemical structure improves the anti-cancer activity/ which functional group of chitosan regulates anti-cancer activity as one Figure. All the above Figures will definitely give significant value to this review and attract more readers.
I have drawn in figure 6 page 20.
New comment: Please provide the reference for Fig.6, Where did you get the concept for Fig.6? This Figure describes the general mechanism of anticancer, provide the specific pathways how chitosan work on cancer cells and which receptor-ligand binding triggers cell signals? How cell receptor is activated by chitosan in breast anticancer mechanism with appropriate evidence using citation.
Don’t include Figure in conclusion section, delete and mention Figure 6 in running text.
- Table 2. Studies on Chitosan-based Nanoparticles of Targeted Drug Delivery Systems.: Describe the available actual mechanism of the ChNPs on the cancer cells listed in Table 2.
The actual mechanism, i have describe in general in page 16-18 below the table 2.
New comment: Yes that is the problem, you have mentioned the mechanism in general. You should describe the actual mechanism of Chitosan in Breast cancer, since this review is mainly focusing on breast cancer, so it is important to describe the specific mechanism in breast cancer. Advice the authors to refer more papers on breast cancer.
4. Conclusion: Not at all related to or focused on breast cancer and ChNPs. This section was also written in a general way. Describing very basic statements regarding the use of ChNPs in cancer treatment, which was already published in several review papers.
Done, i have revised the manuscript
New comment: Remove Fig.6 from conclusion part and mention in running text.
Round 3
Reviewer 2 Report
The revised version is satisfactory and complete.